# Knockdown of DOM/Tip60 Complex Subunits Impairs Male Meiosis of *Drosophila melanogaster*

**DOI:** 10.3390/cells12101348

**Published:** 2023-05-09

**Authors:** Yuri Prozzillo, Gaia Fattorini, Diego Ferreri, Manuela Leo, Patrizio Dimitri, Giovanni Messina

**Affiliations:** 1Department of Biology and Biotechnology “Charles Darwin”, Sapienza University of Rome, 00185 Rome, Italy; yuri.prozzillo@uniroma1.it (Y.P.);; 2Institute of Molecular Biology and Pathology (IBPM), Consiglio Nazionale delle Ricerche (CNR), Sapienza University of Rome, 00185 Rome, Italy; 3Department of Sciences and Technologies, University of Sannio, 82100 Benevento, Italy; 4Pasteur Institute, Fondazione Cenci-Bolognetti, 00161 Rome, Italy; 5Department of Biotechnology and Biosciences, Milano-Bicocca University, 20126 Milan, Italy

**Keywords:** TIP60, DOMINO, ATPase, *Drosophila* male meiosis, chromatin remodeling, cell division, epigenetics

## Abstract

ATP-dependent chromatin remodeling complexes are involved in nucleosome sliding and eviction and/or the incorporation of histone variants into chromatin to facilitate several cellular and biological processes, including DNA transcription, replication and repair. The DOM/TIP60 chromatin remodeling complex of *Drosophila melanogaster* contains 18 subunits, including the DOMINO (DOM), an ATPase that catalyzes the exchange of the canonical H2A with its variant (H2A.V), and TIP60, a lysine-acetyltransferase that acetylates H4, H2A and H2A.V histones. In recent decades, experimental evidence has shown that ATP-dependent chromatin remodeling factors, in addition to their role in chromatin organization, have a functional relevance in cell division. In particular, emerging studies suggested the direct roles of ATP-dependent chromatin remodeling complex subunits in controlling mitosis and cytokinesis in both humans and *D. melanogaster.* However, little is known about their possible involvement during meiosis. The results of this work show that the knockdown of 12 of DOM/TIP60 complex subunits generates cell division defects that, in turn, cause total/partial sterility in *Drosophila* males, providing new insights into the functions of chromatin remodelers in cell division control during gametogenesis.

## 1. Introduction

ATP-dependent chromatin remodeling complexes use ATP hydrolysis to carry out the sliding and eviction of nucleosomes for the incorporation of histone variants [1,2]. These events are required for several cellular and biological processes, including DNA transcription, replication and repair [3].

In *Drosophila melanogaster*, the DOM/TIP60 chromatin remodeling complex belongs to the INO80 family, and it contains 18 subunits that provide a multitude of functions [1,4]. Major subunits of the DOM/TIP60 complex are DOMINO (DOM), an ATPase of the SWI2/SNF2 type that catalyzes the chromatin remodeling activity exchanging canonical H2A with its H2A variant (H2A.V), and TIP60, a lysine-acetyltransferase that acetylates H4, H2A and H2A.V histones [5,6,7,8,9,10]. In addition, the *D. melanogaster* DOM/TIP60 complex includes ACT87E, BAP55, BRD8, DOMINO (A and B), DMAP1, EAF6, E(PC), GAS41, ING3, MRG15, MRGBP, NIPPED-A, PONTIN, REPTIN and YL1 subunits [11,12,13].

In recent decades, several lines of evidence have indicated that ATP-dependent chromatin remodeling factors, in addition to their role in chromatin regulation, have a functional relevance in mitotic cell division [14,15,16,17,18,19,20,21,22,23]. A genome-wide RNAi screening in *D. melanogaster* Schneider 2 (S2) cells identified multiple members of several chromatin remodeling complexes as potential novel regulators of the cell cycle. Specifically, DOM and YETI have been found near APC/C (anaphase-promoting complex/cyclosome) in a mitotic index-based network, suggesting that these components interact with each other [24]. Accordingly, the in vivo expression of CFDP1, the human ortholog of YETI, in *D. melanogaster* determines the formation of inactive heterodimers producing a strong dominant negative effect which affects cell proliferation and differentiation [24]. In the opposite way, the expression of YETI in HeLa cells decreases the mitotic index by impairing cell cycle progression [25,26]. Additional evidence has shown that RNAi-mediated depletion of BAP55 increases the incidence of multinucleated cells and results in spindle assembly defects [27]. More recently, DOM-A and MRG15 [21,22,28] were found to localize on both centrosomes and the midbody, and their depletion leads to mitotic and cytokinesis defects in *D. melanogaster* S2 cells [21]. Similarly, TIP60 and YETI were found in the midbody too, and their knockdown led to cytokinetic failure [22]. Moreover, the *Domino* gene products (DOM-A and DOM-B) are required for normal asymmetric neuroblast (NB) division, and they contribute, together with other TIP60 complex subunits, to NB maintenance and polarity [29]. Taken together, this evidence suggests a specific role for chromatin regulators during mitosis.

In contrast, little is known about the involvement of the TIP60 chromatin remodeling complex during meiosis and spermatogenesis. Indeed, sporadic studies indicated an involvement of DOM/TIP60 complex subunits in meiotic cell divisions. Cenci, et al. [30] have reported the failure of cytokinesis in meiotic cells of *Yeti* mutants. It has also been found that REPTIN and PONTIN function as dynein cytoplasmic assembly factors, highlighting their role in sperm motility and male fertility of *D. melanogaster* [31]. Moreover, E(PC) and NIPPED-A were found to play crucial roles during the early stages of fly germ cell development [32,33]. In particular, E(PC) promotes the mitosis-to-meiosis transition in *D. melanogaster* male germline lineage [34].

The DOM/TIP60 complex was also identified as a regulator in double-strand breaks (DSBs) repair response during oocytes development [35]. This process involves dynamic changes in chromatin structure with a continuous H2A.V/γH2A.V turnover at the break site until repair is finished. A complete absence of H2A.V, both phosphorylated and unphosphorylated, has been observed in MRG15 mutant germline clones, suggesting that the MRG15 subunit is required for γH2A.V exchange during the meiotic prophase [36]. Thus, a meiotic role for the DOM/TIP60 chromatin remodeling complex is worth exploring.

To deepen our understanding of the roles played by the DOM/TIP60 complex in *D. melanogaster* meiosis, the subcellular distribution of DOM/TIP60 complex subunits was studied. We found that the subunits under investigation (BAP55::HA, DMAP1::HA, DOMINO-A, MRG15, TIP60::HA, YETI, YL1::GFP) were localized to sites of the meiotic apparatus. Most notably, the downregulation of 12 subunits (BAP55, DOMINO, DMAP1, EAF6, E(PC), GAS41, MRG15, MRGBP, PONTIN, REPTIN, YETI and YL1) affected chromosome segregation and cytokinesis. Based on these results, we hypothesized that the DOM/TIP60 complex plays crucial roles in different steps of male meiosis progression.

## 2. Materials and Methods

### 2.1. Fly stocks and Genetics

All the stocks used in this work (Appendix A) were raised on standard Bloomington formulation Drosophila medium at 25 °C. *w^1^; P{w^+^, Ubq11>EGFP::alphaTub84B}, P{w[+mC]=His2Av-mRFP1}; P{w^1^, bamP>GAL4VP16}* (shortened *w^1^; EGFP::αTub, H2A.V::mRFP; bam>Gal4*) and *w^1^; P{w^+^, Ubq11>EGFP::alphaTub84B}, P{Ubi-RFP-spd-2}; P{w^+^, bamP>GAL4VP16}* (shortened *w^1^; EGFP::αTub, Spd2::mRFP; bam>Gal4*) were obtained by genetic recombination crosses between stocks carrying a single transgene (Appendix A).

### 2.2. Experimental System: Drosophila Male Meiosis

In *D. melanogaster*, sperm cell production initiates from the asymmetric cell division of gonial stem cells (GSCs), generating a self-renewed GSC and a gonialblast cell (GC), which undergoes four-round mitosis as the transit-amplifying spermatogonia. After mitosis, 16 interconnected germ cells enter meiosis with a prolonged G2-phase as spermatocytes, followed by two rounds of meiotic divisions, in which, first, homologous chromosomes (meiosis I) and, later, sister chromatids (meiosis II) are segregated [37,38,39,40]. Haploid cells (called spermatids) subsequentially undergo nuclear elongation, compaction–protamination and individualization (Figure 1). The challenge of meiosis is then to segregate both homologous chromosomes and sister chromatids accurately so that each gamete receives exactly one copy of each chromosome. When meiotic chromosomes do not properly segregate, aneuploid gametes are generated, which can led to the formation of inviable or poorly viable zygotes due to gene dosage imbalance [40].

### 2.3. Expression of Tagged Remodelers Using the UAS/Gal4 System

*Drosophila* stocks for expressing BAP55::HA, DMAP1::HA and Tip60::HA proteins (UAS>HA-tagged) were purchased from FlyORF [41,42]. Virgin females carrying the UAS>[remodeler]::HA construct were crossed with males carrying the αTubulin84B>GAL4-VP16/TSTL, CyO:TM6B, Tb ubiquitous driver to trigger the expression of the HA-fused protein in all tissues. As negative control, virgin females w^1^ were crossed with males from the same driver stocks (αTubulin84B>GAL4-VP16/TSTL, CyO:TM6B, Tb) and males with only the driver were taken into consideration for the analysis. Testes from generation F1 of adult males expressing the HA-fused remodeler (UAS>[remodeler]::HA/αTubulin84B>Gal4) or not (negative control) were dissected and fixed for immunofluorescence analysis using anti-HA antibodies. The RNAi-mediated downregulation of the targeted proteins in testes was performed with the bag of marbles-GAL4 (bam-GAL4), a spermatogonial-specific driver [41,43]. In the case of UAS-Domino RNAi transgene (VDRC line 7787), the expressed shRNA induces the simultaneous silencing of both *Domino* transcripts coding for DOM-A and DOM-B isoforms.

### 2.4. Cytological Analyses and Immunofluorescence

Testes of 1-day-old adult males were dissected in TIB (testis isolation buffer): 183 mM KCl, 47 mM NaCl, 10 mM Tris pH 6.8.

Spermatids of fresh testes were analyzed by a phase contrast objective of a Nikon Eclipse 50i epifluorescence microscope to evaluate defects in cytokinesis.

For fixed preparations, testes were placed in 8 μL of TIB and squashed on a slide, overlaid with a coverslip, and frozen in liquid nitrogen. Tissue was dehydrated in cold ethanol for 10′, then fixed for 7′ in 4% paraformaldehyde in phosphate-buffered saline (PBS). Fixation was followed by a 30′ PBTx-DOC (0.3% Triton-X and 0.3% sodium deoxycholate in PBS) permeabilization, 10′ staining with 1 μg/mL DAPI (4,6-diamidino-2-phenylindole) in PBS and mounting in anti-fade medium (DABCO, Sigma). For immunofluorescence analyses, after permeabilization, slides were incubated with blocking solution (0.1% Triton X-100 and 5% FBS in PBS) for 30′ at room temperature in a moist chamber, incubated overnight with primary antibodies dilutions (Appendix A) at 4 °C and then washed and incubated with the secondary antibodies for 1h at room temperature in a moist chamber. Nuclei were stained with DAPI as described above. Testes preparations from a minimum of 3 controls and 3 RNAi-induced independent experiments were examined for each assay. Fluorescent 16-cell cysts and spermatids were observed with a Nikon Eclipse 50i epifluorescence microscope equipped with a CCD camera. Images were acquired with NIS-Elements software, provided by Nikon, and processed using Adobe Photoshop (Adobe Systems, Mountain View, CA, USA) and ImageJ software (http://rsbweb.nih.gov/ij/ accessed on 1 March 2022).

### 2.5. Statistical Analysis

Data analyses were performed using the GraphPad Prism software (GraphPad Software, Inc., La Jolla, CA, USA). All results are expressed as mean ± SD values from three independent replicate experiments. A *p* value of less than 0.05 (* *p* < 0.05, compared with the control group) using two-tailed Fisher’s exact test was considered to be statistically significant.

### 2.6. Bioinformatic Analysis

Pairwise alignment of sequences, obtained from UniProt [43], was performed using EMBOSS Needle [44]. Identification, annotation and graphic output of protein domain were performed using SMART [45] and DOG 2.0 [46].

## 3. Results

### 3.1. The Subunits of DOM/TIP60 Chromatin Remodeling Complex Localize to the Meiotic Apparatus

Using immunofluorescence microscopy (IFM), we investigate the subcellular localization of seven subunits (BAP55, DMAP1, DOM-A, MRG15, TIP60, YETI, and YL1) of the DOM/TIP60 chromatin remodeling complex in the meiotic cell division of *D. melanogaster* testes.

Using specific antibodies, we have found that DOM-A and YETI appear to localize along the microtubular structure of the spindle (Figure 2B), while MRG15 shows centrosomal localization (Figure 2B). To depict the meiotic localization of BAP55, DMAP1 and TIP60 for which no antibodies were available, the expression of HA-fused transgenes (lines available from ORFeome stock center, Zurich) was induced by using the UAS-GAL4 binary expression system (see Section 2) [42,47,48].

As shown in Figure 2C, the signal of BAP55::HA overlaps with the spindle (αTubulin), while that of the DMAP1::HA fusion protein was found to overlap with centrosomal structures. InsteadTip60::HA fusion protein localizes along the meiotic spindle. Any specific signal is lost in the negative control. To investigate the localization of the YL1 subunit, we used a stock expressing GFP::YL1-tagged protein under the native promoter. Using anti-GFP antibodies (Appendix A), the GFP::YL1 signal showed a centrosomal localization, which is absent in the w^*^ negative controls (Figure 2D). Taken together, these data suggest a collective relocation of four DOM/Tip60 remodeling subunits from chromatin to the meiotic apparatus.

### 3.2. RNAi Depletion of DOM/TIP60 Complex Subunits

To study the knockdown effects of DOM/TIP60 chromatin remodeling complex subunits in male meiosis, specific RNA interference was activated by bag of marbles-GAL4 (bam-GAL4), a spermatogonial-specific driver [43,44]. The eGFP::αTubulin and mRFP::H2A.V transgenes were used to fluorescently mark spindles and chromatin, respectively. Specifically, eGFP::αTubulin, mRFP::H2A.V; bam˃Gal4 virgin females were crossed to males from 12 RNAi stocks carrying interfering short-harpin to downregulate the following DOM/TIP60 subunits: BAP55, DMAP1, DOMINO, E(PC), EAF6, GAS41, MRG15, PONTIN, REPTIN, TIP60, YETI and YL1 (Appendix A). The F1 progeny of these crosses was analyzed for chromosome segregation and chromatin integrity. The efficiency of RNAi constructs was already checked by semi-quantitative PCR [4]. Following RNAi-mediated depletion of the subunits under investigation induced in testes, we found a category of defects that can include both condensation and chromosome segregation issues with loss of chromatin fragments (Figure 3A,B and Table 1), which are undistinguishable at this level. Thus, we described these defects as chromatin integrity defects (CID). In agreement with the canonical function of the DOM/TIP60 chromatin remodeling complex, we also found that the knockdown of some subunits (BAP55, DMAP1, EAF6, PONTIN, REPTIN, TIP60 and YETI) generates H2A.V mislocalization (HM) with the red (H2A.V) signal, which digresses from the DAPI (YETI and EAF6) or completely disappears (PONTIN), as shown in Figure 3C,D and Table 2. Similar defects were shown as a consequence of a loss of YETI protein during mitosis [30].

Taken together, these data suggested a role of DOM/TIP60 remodeling subunits in maintaining genome integrity not only in the interphase but also during meiotic division.

### 3.3. Spindle Defects

To further explore the possibility that DOM/TIP60 complex subunits could have a specific role in meiotic division, we analyzed young RNAi-knockdown males harboring eGFP::αTubulin, mRFP::H2A.V. As shown in Figure 4C,D and Table 3, the RNAi knockdown of DOM/Tip60 complex subunits leads to alterations in spindle morphology, with the exception of BAP55 RNAi. In particular, the abnormal spindle defects can either be milder, as shown for DMAP1 RNAi, in which the spindle structure is affected but a centrosomal signal is still perceptible, or stronger, as shown for GAS41 RNAi, in which both spindle fibers and centrosomes are no longer distinguishable. In addition, in some cases (DOMINO, GAS41, MRG15, PONTIN, REPTIN, TIP60 and YL1), cells exhibited more of two spindle poles.

To investigate the occurrence of multipolar spindle defects as a consequence of RNAi-knockdown of DOM/TIP60 subunits, we have recombined the Spd2::mRFP centrosomal fluorescent marker with a spindle marker (EGFP::αTub) and coupled it with the Bam>Gal4 driver to achieve RNAi of DOM/TIP60 complex subunits in testes. Then, female *w*; EGFP::αTub, Spd2::mRFP; bam>Gal4* were crossed to homozygous UAS>RNAi males, and the testes of F1 progeny, carrying the three transgenes, were analyzed. As shown in Figure 4C,D and Table 4, the analysis of the squash preparations’ knockdown showed the occurrence of a high percentage of multipolar meiotic spindles in DOM-, GAS41-, MRG15-, PONTIN-, REPTIN-, TIP60- and YL1-depleted testis. Taken together, the results of these experiments suggested a requirement of DOM/TIP60 complex subunits in the maintenance of a proper spindle structure during meiotic division.

### 3.4. Cytokinesis Defects

We extended our analysis to cytokinesis, the crucial step in cell division giving rise to the two daughter cells following the final cut of the cytoplasmic bridge. Homozygous UAS>RNAi males were crossed to w^1^; bam>Gal4 females, and fresh testes squash preparations from the F1 progeny were analyzed for cytokinesis defects during the two meiotic divisions using phase-contrast microscopy. As shown in Figure 5A and Table 5, in the control experiment, *wt* spermatids physiologically showed one nucleus and one Nebenkern (mitochondrial derivative) with comparable sizes (Nu/Nk = 1/1). In contrast, in RNAi-treated samples, a significant percentage of spermatids containing two or four nuclei with only a bigger Nebenkern (Nu/Nk = 2/1 or 4/1) was observed. In most cases, the aberrant ratio found was Nu/Nk = 2/1, suggesting that defective cytokinesis occur.

## 4. Discussion

We have recently found that subunits of SRCAP and p400/Tip60 complexes are recruited to the mitotic apparatus in HeLa and MRC5 cells to ensure proper cell division [21,22]. Similarly, we have found that DOM-A, MRG15, TIP60 and YETI, four subunits of the evolution-related DOM/TIP60 complex, are recruited to the mitotic apparatus and midbody, with their depletion affecting both mitosis and cytokinesis in *D. melanogaster* S2 cells [21,22].

Here, we provided evidence that a similar phenomenon may also occur in vivo during meiotic divisions in *D. melanogaster* males.

Previous studies have reported that *Drosophila* H2A.V is essential to maintain chromosomal structure during mitosis, and it might be also involved in both chromosome segregation and the organization of kinetochore-driven k-fibers [49]. These features are in line with the chromatin integrity defects shown in Figure 3. In fact, the knock-down of BAP55, DMAP1, EAF6, PONTIN, REPTIN, TIP60 and YETI impaired H2A.V localization in spermatocytes. However, while DOM-B and DOM-A isoforms play roles in H2A.V incorporation and removal, respectively, during *D. melanogaster* oogenesis [49], in our experiments, their depletion did not affect the H2A.V localization during male meiosis [49]. On the other hand, DOM-B and DOM-A seem to be crucial to prevent specific cell division defects such as abnormal spindle morphology, multipolar spindles and multinucleated cells (Figure 3, Figure 4 and Figure 5/Table 1, Table 2, Table 3, Table 4 and Table 5). While multipolar spindle formation was only observed following the depletion of a subset of subunits, abnormal spindle morphology and cytokinesis defects were found for all the tested subunits.

The different extent and quality of defects observed following depletion of the tested subunits may be ascribed to the different efficiencies of the siRNA lines used in this work. It is also possible that the lack of certain subunits may have a milder impact on meiotic division compared to others.

Together, these results suggested that the entire complex plays a crucial role in meiotic spindle assembly, possibly participating to microtubule polymerization and/or stabilization. Moreover, the observed meiotic defects strongly suggest that most of the subunits of the DOM/TIP60 remodeling complex can cooperate in the control of several steps of meiotic cell division, possibly maintaining some of their interactions during their relocation, supporting the hypothesis that a number of DOM/TIP60 subcomplexes could assemble to modulate meiosis. Spermatogenesis is a finely regulated process generating highly polarized motile sperms (1.8 mm long) from small round cells (approximately 12 μm in diameter). This process can be dramatically affected by the failure of meiotic divisions. Our preliminary data suggest that RNAi depletion of some DOM/TIP60 complex subunits negatively impacts the physiological elongation of *D. melanogaster* sperms, thus affecting male fertility (Table 6). This matter gains importance in the light of the high level of sequence similarity between analyzed subunits of the DOM/TIP60 complex in *D. melanogaster* and humans (Figure 6, Table 7 and Appendix A). From this perspective, haploinsufficiency mutation of human TIP60 subunits can not only predispose to genetic instability and cancer onset [21,22], but may also affect meiosis and gametogenesis, thus reducing individual fertility.

## 5. Conclusions

In summary, these results highlight the intriguing possibility that subunits of the DOM/TIP60 complex, apart from their canonical functions in chromatin regulation, can maintain their interaction during their relocation to the meiotic apparatus and play essential roles in meiotic cell division to regulate cell cycle progression, centrosome function and spindle organization/function.

Collectively, our results lead to conclude that the subunits of the DOM/TIP60 complex perform extra-chromatin functions not only in mitosis, as shown by Messina et al. [22], but also during the meiotic cell division. In this view, cell division dysfunctions, cancer and infertility appear to be closely interlinked, an aspect that deserves to be further explored by future studies.

## Figures and Tables

**Figure 1 cells-12-01348-f001:**
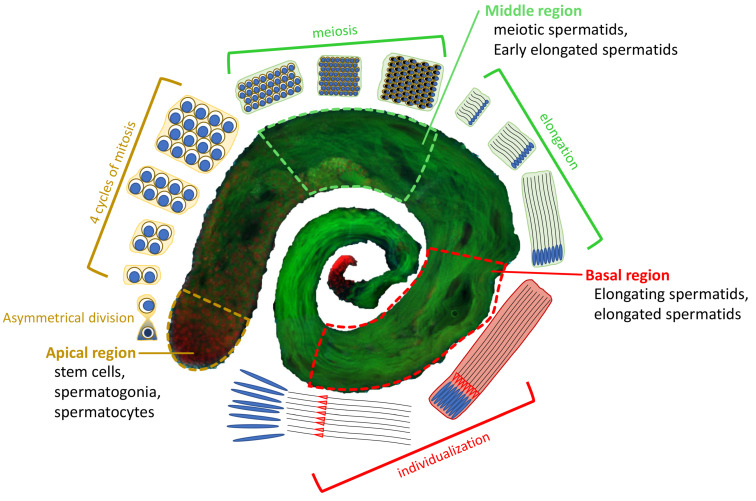
Overview of *D. melanogaster* spermatogenesis. At the apical region of a testis, a hub of germline stem cells (GSCs) divides asymmetrically into two cells: a daughter stem cell and a differentiating gonialblast (GB). The GB goes through 4 cycles of mitosis to form a cyst of 16 primary spermatocytes. Primary spermatocytes will proceed through meiosis, resulting in the generation of 64 roundish haploid spermatids (Middle region). The latter undergo elongation processes characterized by changes in nuclear shape and chromatin condensation to form individualized mature sperm (Basal region), which are stored in the seminal vesicle until fertilization. EGFP::α-Tubulin84D in green, and H2A.V::mRFP in red.

**Figure 2 cells-12-01348-f002:**
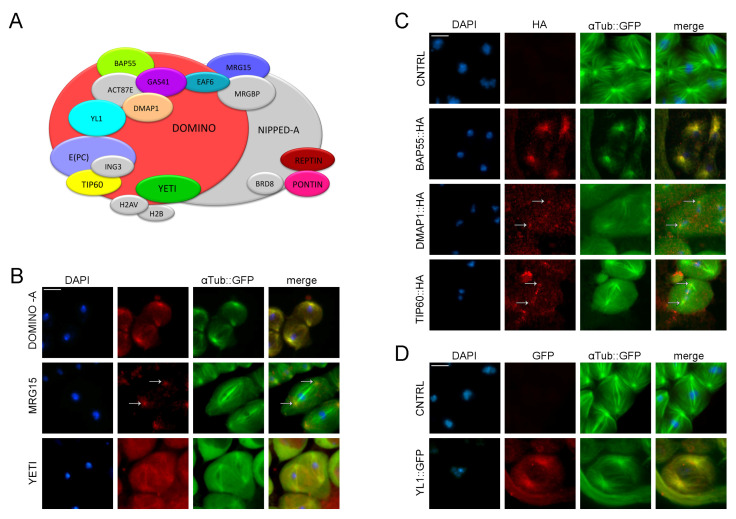
Dynamic localization of DOM/Tip60 chromatin remodeling complex subunits in meiosis. (**A**) Graphical representation of DOMINO/DOM/Tip60 remodeling complex. Subunits are not in scale. (**B**–**D**) Immuno-localization of DOMINO-A, MRG15, Yeti, BAP55, DMAP1, TIP60 and YL1. Testes of young adult, 1–3 days old, from EGFP::αTub; Bam>Gal4 crossed with UAS>protein-HA tag and UAS>protein-GFP tag were stained with specific-antibody (in red) while αTubulin is endogenously fluorescent (in green). DNA is stained with DAPI (in blue). DOMINO-A, MRG15, BAP55, DMAP1 and YL1 showed a specific localization to centrosomes; YETI showed a spindle localization, while Tip60 showed a signal along the meiotic spindle. Scale bar, 10 µm.

**Figure 3 cells-12-01348-f003:**
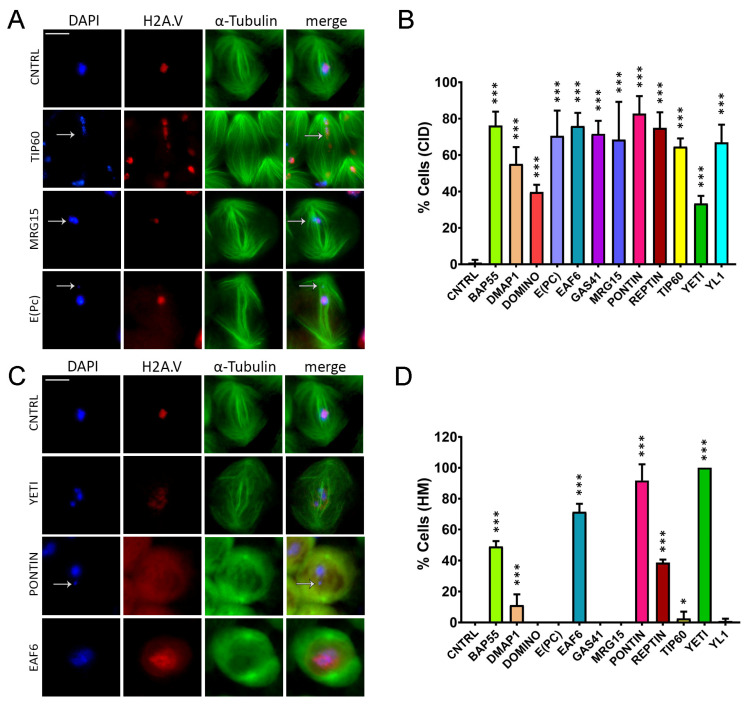
Chromatin integrity defects (CID) and H2A.V mislocalization (HM) defects induced by RNAi in meiosis. Cytological analysis of testis squashes prepared from EGFP::αTub, H2A.V::mRFP; Bam>Gal4 driver crossed with specific remodelers of the RNAi construct. DNA is stained with DAPI (in blue), EGFP::αTub (in green) and H2A.V::mRFP (in red). Scale bar, 10 µm. (**A**) For Tip60, MRG15 and E(pc), the white arrow indicates chromatin fragments probably lost during segregation. (**B**) Quantification analysis of CID after RNAi knockdown effects activated by the EGFP::αTub, H2A.V::mRFP; Bam>Gal4 driver. n = number of analyzed cells: CNTRL (n = 72), BAP55 (n = 128), DMAP1 (n = 260), DOMINO (n = 98), E(PC) (n = 113), EAF6 (n = 289), GAS41 (n = 196), MRG15 (n = 82), PONTIN (n = 281), REPTIN (n = 339), TIP60 (n = 116), YETI (n = 79) and YL1 (n = 102). (**C**) H2A.V mislocalization is reported for YETI, PONTIN and EAF6 as a widespread nuclear signal compared to Control sample. (**D**) Quantification analysis of HM after RNAi knockdown effects activated by the EGFP::αTub, H2A.V::mRFP; Bam>Gal4 driver. n = number of analyzed cells: CNTRL (n = 148), BAP55 (n = 138), DMAP1 (n = 132), DOMINO (n = 64), E(PC) (n = 124), EAF6 (n = 184), GAS41 (n = 195), MRG15 (n = 70), PONTIN (n = 119), REPTIN (n = 85), TIP60 (n = 43), YETI (n = 154) and YL1 (n = 82). The statistical analysis is performed by using two-tailed Fisher’s exact test (* = *p* value ≤ 0.05, ** = *p* value ≤ 0.005, *** = *p* value ≤ 0.0005).

**Figure 4 cells-12-01348-f004:**
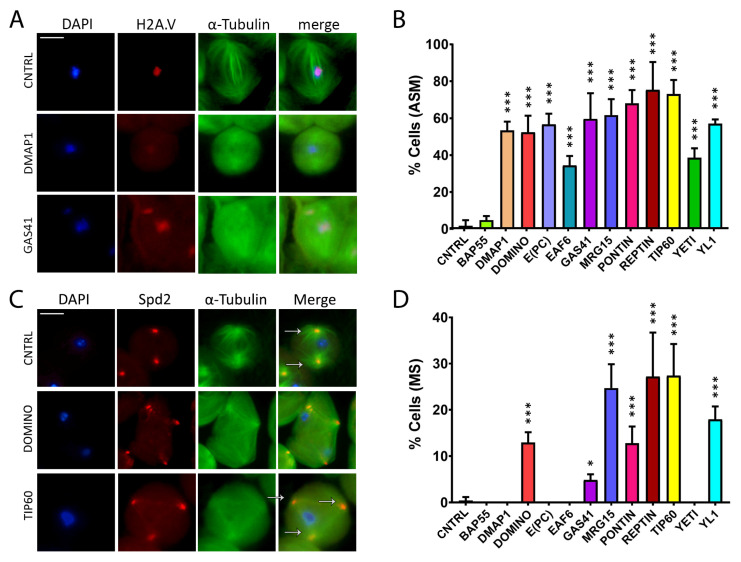
Abnormal spindle morphology (ASM) and multipolar spindle (MS) defects induced by RNAi in meiosis. Testis squashes preparation from Bam>Gal4 driver strain crossed with specific subunit RNAi construct. DNA is stained with DAPI (in blue), EGFP::αTub (in green) and H2A.V::mRFP (in red). Scale bar, 10 µm. (**A**) Early chromatin decondensation effects are shown for DMAP1 knockdown, with a widespread signal for DNA and H2A.V as well, and for GAS41 knockdown, with detached chromatin fragments from the central plate as well. (**B**) Quantification analysis of ASM after RNAi knockdown effects activated by the EGFP::αTub, H2A.V::mRFP; Bam>Gal4 driver. n = number of analyzed cells: CNTRL (n = 85), BAP55 (n = 74), DMAP1 (n = 58), DOMINO (n = 113), E(PC) (n = 28), EAF6 (n = 32), GAS41 (n = 31), MRG15 (n = 68), PONTIN (n = 53), REPTIN (n = 69), TIP60 (n = 65), YETI (n = 87) and YL1 (n = 63). (**C**) Alteration of spindle structure, shown here for DOMINO and TIP60 subunits from squashed testes of young adult flies, 1–3 days old, from EGFP::αTub, Spd2::mRFP; Bam>Gal4 driver crossed with specific subunit RNAi construct. DNA is stained with DAPI (in blue), EGFP::αTub (in green) and Spd2::mRFP (in red). White arrows indicate two centrosomes in the control sample, while in the interfered samples for TIP60 and DOMINO, multiple centrosomes are noticeable. (**D**) Quantification analysis of MS after RNAi knockdown effects activated by the EGFP::αTub, Spd2::mRFP; Bam>Gal4 driver. N = number of analyzed cells: CNTRL (n = 199), BAP55 (n = 131), DMAP1 (n = 246), DOMINO (n = 328), E(PC) (n = 189), EAF6 (n = 147), GAS41 (n = 181), MRG15 (n = 96), PONTIN (n = 119), REPTIN (n = 152), TIP60 (n = 162), YETI (n = 130) and YL1 (n = 192). The statistical analysis is performed by using two-tailed Fisher’s exact test (* = *p* value ≤ 0.05, ** = *p* value ≤ 0.005, *** = *p* value ≤ 0.0005).

**Figure 5 cells-12-01348-f005:**
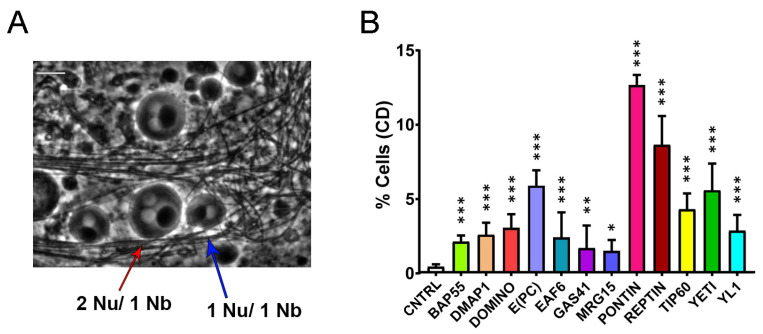
Cytokinesis defects (CD) induced by RNAi-knockdown of DOM/TIP60 complex subunits in meiosis. (**A**) Normal onion-stage cells, in which the ratio between Nucleus (lighter grey circles) and Nebenkern (black circle) is equal to 1:1 with the same volume (blue arrow), while in the case of cytokinesis defects, the ratio become 2:1 or more and the nucleus volume become smaller than the Nebenkern (red arrow). Phase contrast microscopy. Scale bar, 10 µm. (**B**) Quantification analysis of CD after RNAi knockdown effects activated by the Bam>Gal4 driver. n = number of analyzed cells: CNTRL (n = 1231), BAP55 (n = 772), DMAP1 (n = 1327), DOMINO (n = 1061), E(PC) (n = 595), EAF6 (n = 1118), GAS41 (n = 1242), MRG15 (n = 993), PONTIN (n = 922), REPTIN (n = 968), TIP60 (n = 1013), YETI (n = 940) and YL1 (n = 1228). The statistical analysis is performed using two-tailed Fisher’s exact test (* = *p* value ≤ 0.05, ** = *p* value ≤ 0.005, *** = *p* value ≤ 0.0005).

**Figure 6 cells-12-01348-f006:**
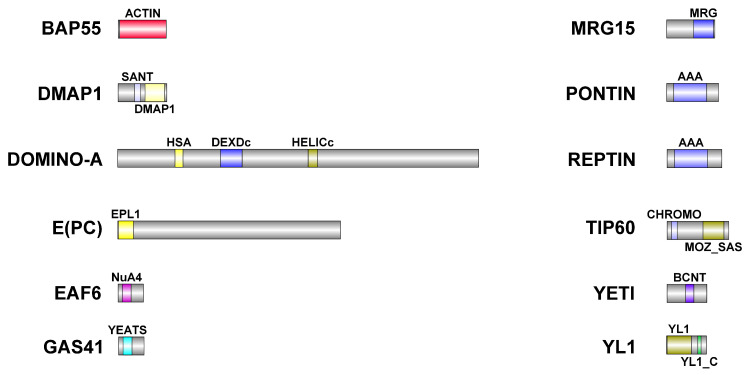
Schematic representation of DOM/TIP60 chromatin remodeling subunits. Functional domains are color-coded and details about their conservation in human orthologues are described in Table 6. Dimensions of proteins and domains are in scale.

**Table 1 cells-12-01348-t001:** Quantification of chromatin integrity defects (CID).

CNTRL	BAP55	DMAP1	DOMINO	E(PC)	EAF6	GAS41
0.90 ± 1.56	76.20 ± 7.64	55.14 ± 9.23	39.69 ± 4.00	70.47 ± 13.93	75.96 ± 7.20	71.59 ± 7.21
	MRG15	PONTIN	REPTIN	TIP60	YETI	YL1
	68.49 ± 20.73	82.80 ± 9.58	74.88 ± 8.63	64.63 ± 4.50	33.37 ± 4.23	66.93 ± 9.64

CID defects are reported as % ± S.D.

**Table 2 cells-12-01348-t002:** H2A.V mislocalization defects (HM).

CNTRL	BAP55	DMAP1	DOMINO	E(PC)	EAF6	GAS41
0 ± 0	0 ± 0	7.69 ± 4.44	0 ± 0	0 ± 0	71.41 ± 5.27	0 ± 0
	MRG15	PONTIN	REPTIN	TIP60	YETI	YL1
	0 ± 0	91.79 ± 10.47	38.66 ± 2.01	2.56 ± 4.44	100 ± 0	0.90 ± 1.56

HM defects are reported as % ± S.D.

**Table 3 cells-12-01348-t003:** Aberrant Spindle Morphology (ASM).

CNTRL	BAP55	DMAP1	DOMINO	E(PC)	EAF6	GAS41
1.75 ± 3.04	4.74 ± 2.26	53.42 ± 4.74	52.37 ± 8.98	56.67 ± 5.77	34.44 ± 5.09	59.60 ± 13.94
	MRG15	PONTIN	REPTIN	TIP60	YETI	YL1
	61.69 ± 8.63	68.14 ± 7.09	75.32 ± 15.10	73.04 ± 7.65	38.53 ± 5.18	56.98 ± 2.36

ASM defects are reported as % ± S.D.

**Table 4 cells-12-01348-t004:** Multipolar Spindle (MS).

CNTRL	BAP55	DMAP1	DOMINO	E(PC)	EAF6	GAS41
0.43 ± 0.74	0 ± 0	0 ± 0	12.93 ± 2.25	0 ± 0	0 ± 0	4.84 ± 1.25
	MRG15	PONTIN	REPTIN	TIP60	YETI	YL1
	24.72 ± 5.14	12.81 ± 3.60	27.18 ± 9.54	27.37 ± 6.88	0 ± 0	17.93 ± 2.81

MS defects are reported as % ± S.D.

**Table 5 cells-12-01348-t005:** Cytokinesis defects (CD).

Control	BAP55	DMAP1	DOMINO	E(PC)	EAF6	GAS41
0.44 ± 0.16	2.13 ± 0.41	2.60 ± 0.81	3.05 ± 0.93	5.89 ± 1.04	2.41 ± 0.63	1.70 ± 1.51
	MRG15	PONTIN	REPTIN	TIP60	YETI	YL1
	1.52 ± 0.72	12.68 ± 0.68	8.64 ± 1.94	4.31 ± 1.07	5.59 ± 1.80	2.87 ± 1.06

CD defects are reported as % ± S.D.

**Table 6 cells-12-01348-t006:** Fertility Test (FT).

CNTRL	BAP55	DMAP1	DOMINO	E(PC)	EAF6	GAS41
100 ± 10.60	77.08 ± 9.59	65.64 ± 9.76 *	71.26 ± 15.13	75.64 ± 7.18	94.78 ± 12.86	63.19 ± 9.19 **
	MRG15	PONTIN	REPTIN	TIP60	YETI	YL1
	79.17 ± 14.31	0 ± 0 ***	0 ± 0 ***	55.42 ± 15.42 **	54.44 ± 2.78 **	56.56 ± 6.23 **

Fertility capability of RNAi males is calculated as percentage ± S.D. with respect to the total progeny generated by CNTRL males. The statistical analysis is performed by using two-tailed Fisher’s exact test (* = *p* value ≤ 0.05, ** = *p* value ≤ 0.005, *** = *p* value ≤ 0.0005).

**Table 7 cells-12-01348-t007:** Conserved domains in analyzed DOM/TIP60 remodeling complex subunits.

*D. melanogaster*	*H. sapiens*	Domain	Identity (%)	Similarity (%)
BAP55 (425aa)	ACTL6A (429aa)	ACTIN	54	71.2
DMAP1 (433aa)	DMAP1 (467aa)	SANT	55.6	79.6
		DMAP1	47.4	68.6
DOMINO-A (3198aa)	SRCAP (3230aa)	HSA	47.2	72.2
		DEXDc	86.1	93.3
		HELICc	91.7	95.2
E(PC) (1974aa)	EPC1 (834aa)	EPL1	1.4	2.8
	EPC2 (807aa)	EPL1	5.7	17.8
EAF6 (225aa)	MEAF6 (191aa)	NuA4	77.5	90
GAS41 (227aa)	YEATS (227aa)	YEATS	77.8	86.4
MRG15 (424aa)	MORF4L1 (362aa)	MRG	51.4	71.8
		Tudor-knot	46.3	59.3
PONTIN (456aa)	RUVBL1 (456aa)	AAA	85.5	92.1
REPTIN (481aa)	RUVBL2 (467aa)	AAA	82.3	93.2
TIP60 (543aa)	KAT5 (513aa)	CHROMO	59.6	82.7
		MOZ_SAS	81.1	86.5
YETI (241aa)	CFDP1 (299aa)	BCNT	49.3	76
YL1 (351aa)	VPS72 (364aa)	YL1	46.6	60.2
		YL1_C	50	70

## Data Availability

The data that support the findings of this study are available from the corresponding author upon reasonable request.

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
