# Peer review of "Knockdown of DOM/Tip60 Complex Subunits Impairs Male Meiosis of Drosophila melanogaster"

_cells, 2023, doi:10.3390/cells12101348_

Round 1

Reviewer 1 Report

The manuscript present evidences of a novel unpredicted role of the chromatin remodeling DOM/TIP60 complex during meiotic divisions. The data are well documented and the issue is definitely of interest. However, the Authors should better clarify some crucial points before the paper could be accepted.

General comments

In the introduction, the Authors stated “We found that the subunits under investigation ….. localized to sites of meiotic apparatus”. However, in the legend of Figure 2 it is clearly reported that the division images refer to mitotic spindles.  

Since in squash preparation the tissue integrity is not maintained and hence the compartimentalization of different cell types become lost, how the Authors distinguished spermatogonial cells (mitotic) from primary spermatocytes (meiotic)?

This clarification is important and necessary since the description of DOM/TIP60 complex subunit pattern in meiotic cells is admittedly more notable than its description in mitotic cells, even though in a tissue never described. Moreover, in the discussion (line 312) the Authors firmly assert “Here we provided evidence that a similar phenomenon [i.e. that DOM/TIP60 complex are recruited to mitotic apparatus and midbody] may also occurs in vivo during the meiotic divisions in D. melanogaster males.”

The Authors should made clear why the DOM/TIP60 complex subunits whose localization they investigated is only a subset of those  whose function they assess by downregulation.

RESULTS section

In either of the Figures a calibration bar is depicted. I think that this type of information is mandatory in a paper in which the role of cytological data is relevant.

Line 164, 

The Authors wrote that they will report the meiotic localization of 12 subunits of the DOM/TIP60 complex, however, they presented the data for only 7 of them. Please explain. 

Fig. 2, it is not clear whether the subunit localization was tested in meiotic cells or in pre-meiotic spermatogonia (see above).

Fig. 2B, the localization of DOM-A appears fully coincident with that of a-tub so the Authors’ claim that it localizes exclusively to centrosomes seems not supported by the image they presented, as instead it is for MRG15. My observation relies also on the pattern in the merge panel, where no green signal (a-tub alone) is apparent but only the yellow color (a-tub + DOM-A, green + red).

Fig. 2C, I note, as I argued above for DOM-A , that the signal of BAP55-HA appears fully coincident with that of a-tub, and not restricted to centrosomes. Moreover, the DMAP1-HA signal appears scattered and not coincident with the centrosomes.

Line 187, “Taken together, these data suggest a collective relocation of four DOM/Tip60 remodeling subunits 188 from chromatin to meiotic apparatus”. This statement raises some interrogatives: why only “four ….. subunits” when the localization of seven is reported? More relevant, how the Authors can speak of a relocation? They showed the localization of DOM/TIP60 complex subunits in only one type of cells (whether meiosis precursor cells, spermatogonia, or meiotic cells is unclear), hence, as such,  the term relocation does not make sense.

Fig. 3. 

1. Again, it is not clear if the images refer to mitotic or meiotic divisions.

2. How chromatin fragments can be distinguished from entire chromosomes? If they were chromosomes, it would be the chromosome alignment in metaphase plate that is impaired (as exemplified by TIP60 depletion). If this was the case, probably it would result from altered chromatin structure at centromeres.

3. H2A.V mislocalization clearly emerges only from PONTIN depletion, being in all the other cases the H2A.V signal coincident with chromatin clumps. 

Table 1. It is clear from a comparison with Figures 3B and D that the reported numbers refer to the percentage of cells showing defects, nevertheless this should be made explicit in the Table heading.

Line 235, remove the full stop after “harbouring” .

Fig. 4C. It is documented that TIP60 silenced flies showed abnormal and multipolar spindles, however in Fig. 3A they exhibited a fully normal spindle. Please explain.

Line 290, “In most cases the aberrant ratio found was Nu/Nk = 2/1, suggesting that defective cytokinesis primarily occurs during the first meiotic division.” I am not convinced of this conclusion.

Since the size of nuclei in the  “Nu/Nk = 2/1” spermatids is approximately double than that in “Nu/Nk = 1/1” spermatids, it must be concluded that the 2/1 ratio arose from the failure of one of the two meiotic (nuclear) divisions. As regards the nebenkern, the presence in abnormal spermatids of a NKs larger than the two associated nuclei would indicate a failure of both meiotic cytokinesis, with the resulting NK four times larger than normal NKs.

DISCUSSION section

Line 314, “Previous studies have been reported that ……… 

Should be rewritten:

Previous studies have been reported that ……..

Line 316, “These defects are in line with chromatin integrity…..” those mentioned are not “defects”, but features or, better, functions.

Line 317, “In fact, the knock-down of BAP55, 317 DMAP1, EAF6, PONTIN, REPTIN, TIP60 and YETI impaired with H2A.V localization in spermatocytes.” Eliminate “with”.

Line 327/28, the Authors hypothesize that the differences in the extent and quality of observed defects could be ascribed to the different efficiencies of silencing. Since the relative efficiency of the different siRNA constructs was reported in Table 2 of Prozzillo et al., 2021, did the Authors investigate a tentative correlation?

Line 335, “possibly maintaining their interactions during their relocation”, this appears contradictory with that documented in Figure 2 where the different complex subunits seems to localize at least to three different sites.

Reviewer 2 Report

uploaded.

Reviewer 3 Report

Overall it is a simple story and provides some interesting insights for the regulation of spermatogenesis. However, some issues should be addressed before considering for publication.

Major issues:

1.     In the Abstract, the background knowledge of DOM/Tip60 complex is presented, but the research results in this manuscript is not mentioned at all. Please summarize the results and add this part into the Abstract.

2.     The knockdown of DOM/Tip60 complex subunits impairs male meiosis of Drosophila. Then is the male reproductive ability altered? Are there mature sperms in the seminal vesicles? It is necessary to add these data.

3.     Authors presented that the meiosis is affected by the knock down of DOM/Tip60 complex subunits. There are several rounds of mitosis before the meiosis in testes. As the bam-Gal4 is a spermatogonial specific driver, then is the mitosis in testes altered by the knockdown of DOM/TIP60 complex in spermatogonia? I would like to know this result.

4.     In the study, the author examined the localizaitons of several subunits of this complex during meiotic cell division. Actually, their localizations are not all same.  In addition, the knockdown of different subunits revealed different severity of mitotic defects. What is the mechanism underlying these phenotypes? The authors thought it is an extra-chromatin function. Do auther have any idea about this extra-chromatin function?

Minor issues:

5. There are quite a few typos/grammatical errors throughout the manuscript such as line 22 (the full stop is missing), line 24 (“D. melanogaster” should be italic as “D. melanogaster”), line 27 (same as line 24). I won’t list every one.

6. In Table 6, the comma in every number should be a point if my understanding is correct. The number, for example, 55,6 should be 55.6. The same issue is seen in P values in the figure legends.

7. Several experimental images, such as Figure 3A (the signals indicated by white arrows), Figure 3C (PONTIN and EAF6), Figure 4A (DMAP1, GAS41), and Figure 4C (DOMINO, TIP60), are not clear. 

8. In line 225, line 254 and line 302-303, Pvalue = 0.05 for significance is very confusing.

9. The control image is lacking for Figure 5A.

Round 2

Reviewer 1 Report

The Authors met near all the criticisms and the manuscript is now suitable for publication.

A last concern regards the interpretation of the Nu/Nk=2/1 ratio.

The Authors propose that “defective cytokinesis primarily occurs in at least one meiotic division”.

I try to explain why this interpretation is flawed.

If the nuclear meiotic divisions proceed regularly, the presence of two nuclei are indicative of the interphase between the first and the second meiotic division. Hence. If this is true, the large Nk would be indicative of a failure of the first cytokinesis. However, this kind of explanation is highly improbable since the interphase between the two meiotic divisions is extremely rapid and in Drosophila there is no checkpoint blocking a defective meiosis. Thus, the Nu/Nk=2/1 ratio could not be the most frequently observed.

The more reliable interpretation of the data is that in the defective 2/1 onion stages, the two nuclei derive from a failure of one of the two nuclear meiotic divisions and the big nebenkern from a failure of both cytokinesis.

If the Authors accepted or not the criticism will not preclude the manuscript publication, however I think they should take into consideration the above reasoning to put forward rigorous and faultless interpretations of their data.